# Comparison of Selected Non-Coding RNAs and Gene Expression Profiles between Common Osteosarcoma Cell Lines

**DOI:** 10.3390/cancers14184533

**Published:** 2022-09-19

**Authors:** Mateusz Sikora, Katarzyna Krajewska, Klaudia Marcinkowska, Anna Raciborska, Rafał Jakub Wiglusz, Agnieszka Śmieszek

**Affiliations:** 1The Department of Experimental Biology, The Faculty of Biology and Animal Science, University of Environmental and Life Sciences Wroclaw, Norwida 27B, 50-375 Wroclaw, Poland; 2Department of Oncology and Surgical Oncology for Children and Youth, Institute of Mother and Child, ul. Kasprzaka 17a, 01-211 Warsaw, Poland; 3Institute of Low Temperature and Structure Research, Polish Academy of Sciences, Okolna 2, 50-422 Wroclaw, Poland; 4Centre for Advanced Materials and Smart Structures, Polish Academy of Sciences, Okolna 2, 50-950 Wroclaw, Poland; 5International Institute of Translational Medicine, Jesionowa 11 St, 55-124 Malin, Poland

**Keywords:** osteosarcoma, microRNA, mRNA, gene expression profile, RT-qPCR

## Abstract

**Simple Summary:**

Osteosarcoma (OS) is a malignant tumour affecting mainly children and elderly people. Despite significant advances in cancer medicine, osteosarcoma patients’ survival is not improving. The primary treatment methods are established using in vitro models that rely upon the application of well-established cell lines, including U-2 OS, Saos-2 and MG-63. The molecular phenotype of these cell lines is still not fully outlined. Therefore, our study aimed to establish the expression profile of molecular markers related to osteosarcoma survival, progression and metastasis. Non-bone-related cells were used as a reference, i.e. HeLa cell line and human adipose-derived stromal cells (hASCs). Evaluated osteosarcoma cell lines showed characteristic phenotypes with unique patterns related to upregulation of MMP-7, MMP-14, BMP-7, miR-21-5p, miR-124-3p and downregulation of lncRNA MEG3. Our findings may facilitate the selection of the most reliable cellular model for pre-clinical investigations focused on developing new and satisfying methods of osteosarcoma therapy.

**Abstract:**

Osteosarcoma (OS) is a bone tumour affecting adolescents and elderly people. Unfortunately, basic treatment methods are still underdeveloped, which has a high impact on the poor survivability of the patients. Studies designed to understand the underlying mechanisms of osteosarcoma development, as well as preclinical investigations aimed at establishing novel therapeutic strategies, rely significantly upon in vitro models, which apply well-established cell lines such as U-2 OS, Saos-2 and MG-63. In this study, the expression of chosen markers associated with tumour progression, metastasis and survival were identified using RT-qPCR. Levels of several onco-miRs (miR-21-5p, miR-124-3p, miR-223-3p and miR-320a-3p) and long non-coding RNA MEG3 were established. The mRNA expression of bone morphogenetic proteins (BMPs), including BMP-2, BMP-3, BMP-4, BMP-6, BMP-7, as well as their receptors: BMPR-IA, BMPR-IB and BMPR-II was also determined. Other tested markers included metalloproteinases, i.e., MMP-7 and MMP-14 and survivin (BIRC5), C-MYC, as well as CYCLIN D (CCND1). The analysis included comparing obtained profiles with transcript levels established for the osteogenic HeLa cell line and human adipose-derived stromal cells (hASCs). The tested OS cell lines were characterised by a cancer-related phenotype, such as increased expression of mRNA for BMP-7, as well as MMP-7 and MMP-14. Osteosarcoma cells differ considerably in miR-21-5p and miR-124-3p levels, which can be related to uncontrolled tumour growth. The comprehensive examination of osteosarcoma transcriptome profiles may facilitate the selection of appropriate cell models for preclinical investigations aimed at the development of new strategies for OS treatment.

## 1. Introduction

Osteosarcoma (OS) is a primary malignant bone tumour formed by cells producing osteoid and immature bone tissue. It mainly affects children and young people (aged 15–19 years) [1]. The second occurrence peak is distinguished in patients over 60–70. Osteosarcoma is located in long bones, such as the femur, humerus or tibia; less often, it affects the skull, jaw or pelvis. Osteosarcoma has highly metastatic potential—invasive behaviour of OS is an adverse prognostic factor for the survival rate. The most successful strategy of osteosarcoma treatment is still chemotherapy, followed by surgical removal of the primary tumour. Such a therapeutic strategy can improve the survival rate to around 70%, while the survival rate for patients treated with surgery alone is 15–17%. Additionally, close attention is paid to modern neoadjuvant chemotherapy and targeted therapies, which makes OS treatment more effective [2]. 

To understand molecular mechanisms underlying osteosarcoma development and to determine the potential usefulness of newly developed therapeutic strategies, scientists design preclinical assays that rely significantly on in vitro models, applying well-established cell lines, including U-2 OS, Saos-2 and MG-63. Thus, this study analysed the molecular characterisation of those acknowledged human osteosarcoma cell lines. The cells are considered good models for functional osteosarcoma studies. However, in the literature, discrepancies in the characteristics of the molecular phenotype of those cells are noted. Pautke et al. profoundly characterized the osteogenic properties of the osteosarcoma cell lines Saos-2, MG-63 and U-2 OS [3]. They tested the expression of main osteogenic biomarkers such as osteocalcin, alkaline phosphatase, decorin and collagens, including collagen type I, type IV and type IX. The study revealed that osteogenic phenotypes differ between osteosarcoma cell lines showing significant heterogeneity. For example, Saos-2 cells are characterized by a mature osteoblast phenotype, while U-2 OS cells do not express most osteoblastic markers, showing more fibroblast-like features. The MG-63 cell line was described as the most heterogenic, with both mature and immature osteoblastic features. 

Furthermore, Fromigue et al. indicated that the highest migratory capacity characterizes Saos-2, while MG-63 cells have the lowest [4]. In turn, the U-2 OS cell line was the most invasive. The molecular phenotype of OS cell lines affects the differentiation potential of OS cell lines, which was shown by Choong et al., who reprogrammed OS cell lines to induce pluripotent cancer cells [5]. The study was performed on Saos-2, MG-63, G-292 and U-2 OS, indicating the highest transduction efficiency in U-2 OS. At the same time, U-2 OS cells were characterized by the lowest reprogramming stability in long-term cultures, which was linked to their high invasiveness and proliferative activity.

Given the differences in the molecular phenotype of OS cell lines, the main objective of this paper was to investigate the expression profiles of several non-coding RNAs and mRNAs of genes involved in bone biology and tumour progression. The expression profiles of selected biomarkers determined in OS cell lines were compared with the HeLa cell line, known for its osteoinductive properties mediated by bone morphogenetic proteins (BMPs) [6,7], as well as with human multipotent adipose tissue-derived stromal cells (hASCs), constituting a tumour microenvironment [8], as well as contributing to osteosarcoma progression, metastasis and proliferation [9].

We have tested levels of so-called onco-microRNAs (onco-miRs), including miR-21-5p, miR-124-3p, miR-223-3p and miR-320a-3p, and long non-coding RNA—MEG3. The onco-miRs regulate multiple cellular processes of cancer cells, including their division, differentiation and viability. For example, functional in vitro studies showed that miR-21-directly targets the tumour suppressor phosphatase and tensin homologue (PTEN), regulating the proliferation of osteosarcoma cells (MG 63 and U-2 OS) and head and neck cancers (HNC) cell lines [10,11]. Additionally, Sheng et al. showed that modulation of the miR-21-PTEN axis affects cisplatin resistance in HNC [11]. Furthermore, miR-124, miR-223 and miR-320a are recognized as osteosarcoma suppressors determining cell migration and invasion in vitro [12,13,14]. Recently it was found that long non-coding RNA MEG3 (lncRNA MEG3) enhances the chemosensitivity of OS associated with antitumor immunity regulated via the miR-21-5p/p53 pathway and autophagy. At the same time, overexpression of lncRNA MEG3 inhibits cell proliferation and migration [15]. Furthermore, Shi et al. showed that overexpression of lncMEG3 promotes spontaneous apoptosis of osteosarcoma cell line MG-63, significantly increasing levels of early and late apoptotic cells [16]. 

Plethora of onco-miRs also regulate the expression of bone morphogenic proteins [17]. Thus, in the present study, we have tested mRNA levels for bone morphogenic proteins, namely BMP-2, BMP-3, BMP-4, BMP-6, BMP-7, and their receptors: BMPR-IA, BMPR-IB and BMPR-II. We have decided to establish the expression profile of the selected biomarkers as they were recognized as essential molecules regulating bone development both in physiological and pathological conditions. Bone morphogenic proteins belong to the transforming growth factor-beta superfamily (TGF-β) and display a pleiotropic effect playing a pivotal role in tissue development during embryogenesis. However, the role of BMPs as morphogens did not restrict bone formation during embryonic development. They are also essential in maintaining adult tissue homeostasis regulating various aspects of progenitor cell biology, including proliferation, differentiation and apoptosis [18,19]. BMPs elicit their biological effects by activating specific combinations of type I and II serine/threonine kinase receptors (BMPR-IA, -IB and BMPR-II). BMP pathways are regulated by numerous non-coding RNAs, including miR-21-5p and miR-320 [20,21]. Other proteins essential for bone extracellular matrix remodelling and regulated by ncRNA are matrix metalloproteinases (MMPs), also overexpressed in malignant tissues and considered markers of tumour aggressiveness and metastatic potential, indicating poor prognosis [22]. Thus, in this study, we determined the expression of metalloproteinases, i.e., MMP-7 and MMP-14 involved in paediatric tumour pathogenesis, including osteosarcoma [23].

Moreover, given that miRNAs regulate genes controlling apoptosis and cell cycle arrest, we decided to determine mRNA levels of several pro-survival markers highly expressed in various malignant tumour tissues and tumour cell lines, including osteosarcomas. The prominent anti-apoptotic molecule, i.e., survivin (BIRC5), was identified as a gene highly expressed in osteosarcoma tissues and involved in osteosarcoma development [24]. C-MYC, a proto-oncogene belonging to the Myc family, is also overexpressed in OS cell lines, including MG-63 and 143B [25,26]. Moreover, it was shown that miRNAs regulate the expression of cyclin D1, a key molecule contributing to the proliferation of osteosarcoma cells [27].

A functional characteristic of osteosarcoma based on ncRNA and mRNA profiles is crucial for the proper selection of representative models and in seeking suitable candidates for future therapeutic targets [17,18]. The cell lines used in the study are relevant models for preclinical studies aimed at developing novel potential therapeutic strategies. Thus, the overall aim was to determine the expression pattern of transcripts potentially associated with bone tumour progression and metastasis. The proposed panel includes analysis of several onco-miRs, i.e., miR-21-5p, miR-124-3p, miR-223-3p and miR-320a-3p and lncRNA MEG3, as well as detection of mRNA for BMP-2, BMP-3, BMP-4, BMP-6, BMP-7, BMPR-IA, BMPR-IB, BMPR-II, MMP-7, MMP-14, survivin (BIRC5), C-MYC and CYCLIN D (CCND1).

## 2. Results

The analysis of non-coding RNAs (Figure 1) revealed that miR-21-5p (miR-21) is highly expressed in osteosarcoma cell lines, and the levels of miRNA-21 transcripts are significantly increased compared to HeLa cell line and stromal cells, i.e., hASC. The highest levels of miR-21 were noted in U-2 OS and MG-63 cells (Figure 1a). Significantly increased levels of miR-124-3p (miR-124) also characterised those two OS cell lines. In turn, the expression of miR-124 was comparable between Saos-2 and hASCs, while the lowest levels of miR-124 transcripts were noted in the HeLa cell line (Figure 1b). No significant differences between miR-223-3p (miR-223) and miR-320a-3p (miR-320a) levels were found between osteosarcoma cell lines and hASC. The lowest accumulation of tested miRNAs was noted in the HeLa cell line (Figure 1c,d). Moreover, the analysis showed that lncRNA MEG3 expression is significantly lowered in OS cells, while its levels are increased in the cancerous HeLa cell line and stromal hASC. The lncRNA MEG3 levels in HeLa and hASC were comparable (Figure 1d).

Analysis of mRNA levels for particular BMPs showed common gene expression profiles of BMP-2 and BMP-6 molecules (Figure 2a,d). Increased mRNA levels for BMP-2 and BMP-6 characterized U-2 OS and Saos-2 cell lines and were comparable with transcript levels noted in the HeLa cell line. The lowest expression of mRNA for BMP-2 and BMP-6 was indicated in MG-63 cells. The hASC were distinguished by the lowest BMP-2 and BMP-6 transcript levels compared to the tumour cell lines. Moreover, we noted significantly increased levels of BMP-3 transcripts in U-2 OS and Saos-2 cells, while the expression of mRNA for BMP-3 was lowered in MG-63, HeLa and hASC (Figure 2b).

Further, BMP-4 expression was significantly increased in Saos-2, while mRNA levels for BMP-4 in U-2 OS cells were similar to the levels established in the HeLa cell line. Lowered expression of BMP-4 was indicated in MG-63 and hASC (Figure 2c). On the other hand, all examined OS cell lines showed a high accumulation of *BMP-7* transcripts compared to the HeLa cell line and hASC population (Figure 2e).

The evaluation of mRNA levels for BMPRs (Figure 3) showed that both U-2 OS and Saos-2 are characterized by a significantly increased expression of BMPR-IA and BMPR-IB compared to hASCs (Figure 3a,b). Additionally, mRNA levels for BMPR-IA noted in U-2 OS were markedly higher than those measured in the HeLa cell line. In turn, BMPR-IA levels determined in Saos-2 were comparable to the transcripts accumulated in the HeLa. MG-63 and multipotent hASCs showed the lowest expression of BMPR-IA and BMPR-IB. Increased mRNA levels of BMPR-II were characteristic for Saos-2 cells, while the BMPR-II transcript levels noted in U-2 OS and MG-63 were comparable to those indicated in reference cell lines (HeLa and hASC, Figure 3c). The comparative analysis also revealed that mRNA expression for BMPR-II is higher in hASC than in the HeLa cell line. 

The analysis of mRNA for C-MYC revealed that high levels of these transcripts were detected in two osteosarcoma cell lines, U-2 OS and MG-63. The lowest number of C-MYC transcripts has been noted for Saos-2 and was comparable with the C-MYC mRNA levels determined in hASC. Interestingly, the highest accumulation of C-MYC was noted in HeLa cells (Figure 4a). A significantly increased expression of cyclin D (CCND1) was also found in Saos-2. The lowest accumulation of mRNA for CCND1 was determined in MG-63 and hASC. The CCND1 levels were comparable in U-2 OS and HeLa cell lines (Figure 4b). It was also found that the survivin levels (BIRC-5) are lowered in osteosarcoma cells compared to the HeLa cell line. Multipotent non-cancerous cells, i.e., hASCs displayed the lowest expression of BRIC-5, distinguishing this population from tumour cell lines (Figure 4c). The mRNA levels for metalloproteinases, MMP-7 and MMP-14, were significantly increased in all analysed osteosarcoma cells compared to the HeLa cell line and hASCs (Figure 4d,e). A higher accumulation of MMP-7 transcripts was noted in U-2 OS. The expression of mRNA for MMP-7 in Saos-2 and MG-63 was comparable (Figure 4d). A characteristic pattern of MMP-14 was observed in OS cell lines. The highest expression was found in MG-63, while the lowest was in Saos-2 (Figure 4e). In addition, our studies showed that the HeLa cell line, when compared to hASCs, displays decreased mRNA levels for MMP-7 and increased mRNA levels for MMP-14 (Figure 4d,e). 

The transcriptome profile determined by us in OS cell lines, HeLa and hASC were also assembled as a heat map (Figure 5).

## 3. Discussion

Osteosarcoma is a very aggressive malignant tumour of the bone structure with high chemo- and radiotherapy resistance [28,29]. Several studies have shown that well-studied osteosarcoma cell lines, i.e., U-2 OS, Saos-2 and MG-63 show different molecular phenotypes [3,30,31]. These cells are often used as representative models for studies aimed at understanding osteosarcoma biology or studies addressing novel targeted therapies. In this study, we have determined the expression of molecules involved in OS development, progression and survival. The transcripts were measured in U-2 OS, Saos-2 and MG-63 and compared with levels defined for the HeLa cell line and human adipose-derived stromal cells (hASCs). We decided to use HeLa cells as a reference due to their osteoinductive properties manifested by the induction of heterotopic ossicles in the muscles of immunosuppressed mice [6,7]. In turn, the second reference model, i.e., multipotent hASCs were recently identified as potential promoters of osteosarcoma progression, significantly affecting the tumour microenvironment [32,33]. 

The detailed characterisation of osteosarcoma transcriptome is crucial for identifying potential biomarkers and novel molecular targets for designing effective selective therapies [33,34]. Therefore, the biology of OS cells was also studied extensively using in vitro models. The most commonly used OS cell lines differ in their morphology, phenotype and biology. U-2 OS and Saos-2 present typical epithelial morphology in contrast to MG-63, which resembles a fibroblast-like morphotype [3]. U-2 OS shows a low expression of osteocalcin (OCL) or osteopontin (OPN)—molecules considered late markers of osteogenesis. At the same time, the synthesis of OPN and OCL by Saos-2 cells is increased. Additionally, the activity of ALP was previously detected only in Saos-2, which also confirms the mature osteoblastic profile of these cells [3]. The collected data indicate that a particular OS cell line has a specific and unique phenotype affecting its cellular behaviour [34,35,36]. There is a great need for comprehensive characterisation of OS cell line models, mostly due to limited access to primary patient material associated with intensive treatment regimens. Also, those cells are often used for implantable biomaterials testing due to their osteoblastic phenotype. Given the above, a careful evaluation of their gene expression profile is critical in functional studies [30].

In this study, we determined levels of several miRNAs vital for cancer cell biology, including miR-21, miR-124, miR-223 and miR-320, and long-non-coding RNA MEG3. The results indicate that the levels of all studied miRNAs are increased in the osteosarcoma cell lines, while lncMEG3 is decreased. 

The highest expression of miR-21-5p was noted in the U-2 OS cell line, which is consistent with the results presented by Vanas et al. [37]. Previously, miR-21 was recognized as essential onco-miR, playing a role in OS progression [38,39]. MiR-21 suppresses PTEN expression activating the PI3K/Akt pathway associated with cancer longevity [40]. It was also found that miR-21 levels detected in the serum of osteosarcoma patients are significantly increased compared to healthy control subjects. Further, Hua et al. indicated that the expression level of miR-21 in 65 out of 69 patients with osteosarcoma is considerably higher than that noted in adjacent healthy bone tissues [41].

MiR-21 is not only an essential component for the regulation of cancer cell proliferation but may also determine sensitivity to cisplatin, which is a drug used for osteosarcoma treatment. Vanas et al. noted that the inhibition of miR-21 activity desensitises U-2 OS cells to cisplatin treatment. Moreover, the study showed lowered expression of miR-21 in the Saos-2 cell line, which is inconsistent with our results. These discrepancies in the obtained profiles may result from different methods used to determine miRNA levels. Vanas et al. measured miRNA expression using the Northern blot technique, while we tested miRNA levels using RT-qPCR, which is more sensitive, thus yielding highly accurate transcript levels [42]. 

Huang et al. showed that lncRNA MEG and miR-21-5p share the miRNA response element and that their levels are negatively correlated. Moreover, we have found that increased levels of miR-21 are associated with low expression of lncRNA MEG3. Our study confirmed decreased expression of MEG3 in the osteosarcoma cell lines. The study revealed a potential molecular axis, indicating that MEG3 is a regulator of anti-tumour immunity and chemosensitivity regulating the miR-21-p53 pathway [43]. 

The downregulation of lncRNA MEG3 was noted in patients with osteosarcoma and was associated with tumour progression and metastasis [44]. Moreover, Shi et al. showed that lncRNA MEG3 levels were lowered in the MG63 cell line and Saos-2 cell line compared with a normal human foetal osteoblast cell line, which agrees with our study. In contrast, overexpression of MEG3 in MG63 cells resulted in increased cell apoptosis and inhibition of cell proliferation, invasion and migration [16].

Various studies, including ours, revealed that miR-21 is a molecule regulating the differentiation of osteoblast progenitor cells. The current study found that HeLa, known for its ectopic bone formation ability, showed decreased levels of miR-21 compared to OS cell lines. Similarly, hASC used in this study showed lowered expression of miR-21 when compared to the tested OS cell lines. We previously demonstrated that miR-21 levels increase in response to osteogenic stimuli in pre-osteoblast cells [45,46]. It was also shown that ASC could display increased levels of miR-21, which corresponds with the pro-apoptotic phenotype of the progenitor cells derived from horses with equine metabolic syndrome. The role of miR-21 in the biology of ASC is still being investigated, but recent studies point to its adipogenic effect associated with inhibiting TGF-β receptor 2 (TGFBR2) expression [47].

The molecule miR-124 is also an essential molecule regulating gene expression in bone metabolism. miR-124 overexpression leads to lowered expression of RUNX2 and alkaline phosphatase (ALP) activity. Moreover, miR-124 targets the expression of osterix (SP7/OSX), inhibiting the differentiation of progenitor cells into osteoblasts [48]. It was also found that miR-124 targets B7 homolog 3 (B7-H3) and decreases the proliferation and invasion of osteosarcoma cells. Thus, the five-year overall survival rate estimated for patients with upregulated miR-124 is 61.5%, while patients with lowered miR-124 expression have a poor prognosis, estimated at 11.8% [49]. The study by Cong et al. confirmed the potential prognostic and diagnostic value of miR-124, showing that its serum levels are significantly decreased in osteosarcoma patients in an advanced clinical stage or with positive distant metastasis [50]. Geng et al. showed that miR-124-3p levels are low in osteosarcoma tissue and in U-2 OS, Saos-2 and MG-63 [51]. However, those data were retracted, and their validity needs to be verified. The miR-124 levels determined in osteosarcomas cells were compared with adjacent non-neoplastic tissues. In this study, the comparative analysis was not performed to illustrate differences in the miR-124 expression pattern between cell lines. The results presented by Geng et al. showed comparable levels of miR-124 between U-2 OS and MG-63, which is consistent with the results obtained in our study. Moreover, Geng et al. demonstrated that Saos-2 could be distinguished from other OS cell lines by the highest levels of miR-124. However, our study showed that the lowest levels of miR-124 characterize the Saos-2 cell line. This expression pattern can be associated with a mature osteogenic phenotype of Saos-2 cells and increased expression of several osteogenic markers [52]. For instance, we also demonstrated that Saos-2 cells are characterized by increased expression of BMP-3 and BMP-4. Several studies revealed that miR-124 acts as a tumour suppressor, regulating the proliferation and invasion of OS cells. The proposed target of miR-124 is snail family zinc finger 2 (Snail2) which stimulates angiogenesis, proliferation and invasion in various malignancies, including OS [12]. Moreover, Snail2 promotes the epithelial-mesenchymal transition (EMT) of OS cells and metastasis. Therefore, miR-124 mimics are considered to be potential tool for OS therapy. 

The tested reference cell lines, i.e., HeLa and hASC, showed a different pattern of miR-124 expression. We noted significantly decreased levels of miR-124 in the HeLa cell line, which is consistent with the study by Zhang et al., who indicated that miR-124 levels are reduced in HeLa and other human cervical cancer (CC), namely SiHa. The study by Zhang and his group also demonstrated that the overexpression of miR-124 might suppress proliferation and decrease the migratory and invasive properties of the tested CC cell lines. Increased expression of miR-124 induced in HeLa also inhibited the epithelial-mesenchymal transition process in vitro [53].

Further, we demonstrated that miR-223 levels are comparable between U-2 OS, Saos-2 and MG-63 cells. MiR-223 is proposed as a good prognostic and diagnostic marker in osteosarcoma. Significantly lowered expression of miR-223 was noted in osteosarcoma patients compared to the healthy controls [54]. The study by Dong et al. also indicated that miR-223 levels are lowered in osteosarcoma cell lines compared to a human osteoblast cell line (hFOB). Further, increased levels of miR-223 affect the migratory potential of MG-63 osteosarcoma cells, reducing their invasiveness [54]. The expression profile determined in this study showed that miR-223 levels noted for multipotent hASC are comparable with OS cells. Increased expression of miR-223 found in hASC may be correlated with their adipogenic phenotype and origin [55]. In this study, we also demonstrated that the highly metastatic HeLa cell line is characterised by decreased expression of miR-223. Tang et al. previously indicated that low expression of miR-223 noted in HeLa cells is associated with increased EMT ability. In turn, miR-223 mimics introduced to HeLa cultures inhibit their progression by suppressing the EMT process [56]. 

This observation also corresponds with the expression pattern of miR-320, which is an essential modulator of cancer metastasis and proliferation. Increased cell viability with simultaneous downregulation of miR-320a was noted in various cancers, e.g., breast cancer [57], cervical cancer [58], or glioma [59], as well as multiple cell lines including HeLa. However, the miR-320 function in the biology of OS cells is still being investigated. Wu et al. compared miR-320 levels in OS cell lines and osteoblasts, confirming that miR-320 levels are significantly reduced in U-2 OS, Saos-2 and MG-63. The miR-320 expression profile presented by Wu et al. for OS cell lines is in line with our results, showing that the highest levels of miR-320 are noted in MG-63 cells, while the lowest are in U-2 OS [60]. It was found that a potential target for miR-320 is the E2F1 gene, regulating the cell cycle of osteosarcoma [60] or fatty acid synthase (FASN) involved in OS metastasis [61]. Moreover, it was demonstrated that increased expression of miR-320 is related to decreased levels of DLX5 (distal-less homeobox 5), RUNX2 and OSX, indicating an osteoporotic phenotype of the cells [62]. The miR-320 levels determined in hASC were comparable with the levels established for OS cell lines. Increased expression of miR-320 can be related to adipogenic lineage commitment of multipotent stromal cells [63]. miR-223 and miR-320 produced and secreted by hASCs could be the factors responsible for OS progression and development.

The ASC may promote osteosarcoma growth, invasion and migratory properties by releasing exosomes that carry proteins, mRNA and various non-coding RNAs [9]. ASCs may not only contribute to the formation of a microenvironment promoting tumour development but also develop into OS-like tumours. Accumulating evidence demonstrates that the crosstalk between multipotent stromal cells and osteosarcoma significantly influences the “stemness” of cancer cells; thus, a comparison of their transcriptome will be essential to identify potential targets for efficient cancer therapy.

The analysis of mRNA expression for BMP-2 showed that its levels are similar in U-2 OS, Saos-2 and HeLa cells. A recent study by Tian et al. revealed that BMP-2 activates the Wnt/β-catenin signalling pathway and promotes the EMT of osteosarcoma [26]. Additionally, our results indicated that Saos-2 expresses high mRNA levels for BMP-3 and BMP-4, which is consistent with Raval et al., who demonstrated that the osteogenic potential of Saos-2 is associated with an appropriate combination of expressed BMPs, including BMP-2, -3, -4, -6. Interestingly, the MG-63 cell line was characterised by the lowest accumulation of BMP-4 transcripts. We also found that mRNA levels for BMP-6 between tested OS cell lines were comparable. The role of BMP-6 in osteosarcoma development is not well explored. However, it was shown that high expression of BMP-6 is found in osteosarcoma tissues with chondroid differentiation [64]. Additionally, a good correlation between elevated BMP-6 expression in prostate adenocarcinoma and osteoblastic metastases is noted [65]. 

BMPs are highly expressed in the HeLa cell line, which confirmed the study performed by Kochanowska et al. [7,66]. However, our study showed that BMP-7 could be considered an osteosarcoma-specific marker. A high mRNA expression for BMP-7 was noted in all tested OS cell lines. Previous studies showed that BMP-7 and BMP-8 are overexpressed in osteosarcoma tissues, pointing to the importance of those molecules in tumour development [64]. Lowered mRNA levels of the detected BMPs noted in hASC may be associated with the undifferentiated status, since increased expression of BMPs is found during the osteogenic differentiation of ASC [67]. 

We have also determined the mRNA expression profile of BMP receptors (BMPRs), including BMPR-IA, BMPR-IB and BMPR-II, which initiate intracellular pathways through SMAD proteins (canonical signalling). However, it may also act via the JNK and p38 MAP kinase pathway [20,68]. The highest expression of BMPR-IA was noted in the U-2 OS cell line. The mRNA levels for BMPR-IB and BMPR-II noted in U-2 OS were comparable to those indicated in the HeLa cell line. On the other hand, Saos-2 demonstrated significantly increased levels of BMPR-II, while MG-63 showed the lowest transcript levels for both BMPR-IA and -IB. These studies confirmed that BMPs’ signalling elements are expressed in OS cell lines. Nonspecific expression patterns of BMPs and their receptors were noted previously, both in OS tissues and cell lines. The link between BMPs’ signalling and osteosarcoma prognosis was described as tenuous due to different results obtained and presented in various groups. However, the study by Guo et al. showed that the expression of BMPR-II correlates with metastasis in osteosarcomas and indicated that BMPs could participate in tumour aggressiveness or progression [69]. Furthermore, we observed lowered mRNA expression for BMPR-IA and BMPR-IB in hASCs. We also determined that mRNA expression for BMPR-II was significantly increased in hASC compared to HeLa cells. The BMPR-II was previously found to be elevated in subcutaneous adipose tissue compared to visceral adipose tissue. Moreover, it was found that BMPR-II expression is significantly increased in adipose tissue derived from overweight and obese patients [70,71]. 

Because non-coding RNAs regulate BMPs’ signalling cascade, their correlation has started to be intensively studied. The correlation between miRNA and BMPs can disclose the molecular aetiology of different diseases. For example, it was shown that miR-22 and BMP-7/6 form a regulatory feedback circuit in the course of renal fibrosis [72]. In this axis, miR-22 inhibits BMP-7/6, while miR-22 expression is induced by BMP-7/6. Yang et al. also showed that BMP-2 levels in patients with osteoarthritis (OA) are negatively correlated with the concentrations of miR-22 and positively correlated with miR-140. The TGF/BMP signalling may regulate the biogenesis of microRNAs. This phenomenon was described for miR-21 by Davis et al. They showed that TGFβ and BMP signalling enhances the processing of primary transcripts of miR-21 into precursor miR-21 by the DROSHA complex in human vascular smooth muscle cells [73]. Our studies showed that in all tested OS cell lines, the high levels of miR-21 correlated with a high accumulation of BMP-7 transcripts. This molecular phenotype may indicate the increased proliferative potential of OS cell lines and the anti-apoptotic effect of both miR-21 and BMP-7 shown previously in other cancer models [74,75]. 

In our study, we have also determined the expression of genes strictly related to cancer development and progression. The obtained results indicated that the expression of C-MYC in all OS cell lines is lowered compared to HeLa cells. C-MYC is a transcription factor significant for cancer cell proliferation and invasion. The depletion of C-MYC in HeLa cells and MDA-MB-231 cells (human breast adenocarcinoma) strongly impaired their proliferative and migratory activity, which was also accompanied by decreased accumulation of β-catenin [76]. Moreover, the overexpression of C-MYC in the Saos-2 cell line promoted its invasion and increased its viability [77]. A previous study by Xu et al. indicated that C-MYC is overexpressed in OS cell lines and tissues [78]. The mRNA levels of C-MYC determined by Xu et al. in U-2 OS, Saos-2 and MG-63 were comparable. However, in this study, we demonstrated that Saos-2 expresses the lowest levels of C-MYC transcripts, while MG-63 has the highest accumulation of mRNA for C-MYC. This result is consistent with data pointing to superior MG-63 invasiveness activity compared to Saos-2 [77,79]. An opposite expression pattern was observed in terms of CYCLIN D expression. The highest expression was detected in the Saos-2 cell line, while it was lowered in MG-63. The obtained results correspond with the data presented by Huang et al., who showed that Saos-2 accumulates a higher amount of endogenous CYCLIN D protein and that its expression can be regulated by miR-191 [80]. Moreover, it was demonstrated that miR-124 is highly expressed in MG-63, while Wu et al. showed that the upregulation of miR-124 significantly decreases CYCLIN D expression in the human prostate cancer cell line DU145 [81]. Human osteosarcoma cell lines express different progression patterns through the cell cycle [52] therefore, CYCLIN D levels may vary between U-2 OS, Saos-2 and MG-63.

Moreover, we showed that hASCs were characterised by lowered expression of both C-MYC and CYCLIN D. The study by Melnik et al. demonstrated that the expression of c-MYC protein correlates with increased proliferative activity of ASCs and may significantly decrease during its cultivation [82]. Paula et al. noted an approximately two-fold decrease of C-MYC transcripts in ASCs from passage 4 (p4) to 10 (p10) [83]. Moreover, Melnik et al. also showed that multipotent stromal cells isolated from bone marrow (BMSC), characterised by increased constitutive expression of C-MYC, may display lowered potential for tissue-specific differentiation [82]. It was also revealed that CYCLIN D plays a pivotal role in the proliferation of ASCs. The decreased protein level of CYCLIN D caused a decline in ASCs’ percentage in the G_0_/G_1_ phase and resulted in a shift toward rapid proliferation [84].

Another molecule regulating the division of cells is the survivin belonging to the inhibitor of apoptosis (IAPs) family. Gene coding of the survivin (BIRC5) is upregulated in various malignancies, including osteosarcoma [85]. These results indicated that mRNA for survivin is highly expressed in HeLa cells. However, high mRNA levels for the survivin were also noted in Saos-2 cells, which correlates with downregulated miR-124. It was previously shown that miR-124 mimic-transfected Hela cells were characterized by decreased expression of survivin and increased mRNA for pro-apoptotic BAX levels [86]. We demonstrated a lowered accumulation of survivin transcripts in hASCs. Increased survivin expression is noted in ASCs derived from obese patients and may act as an inhibitor of apoptosis, reducing inflammation and metabolic consequences of obesity. It is also postulated that the survivin levels may be nodal, a protein that regulates the crosstalk between adipose tissue and tumour cells [87].

Finally, our study showed that all tested OS cell lines expressed significantly increased mRNA levels for metalloproteinase 7 and 14 (MMP-7 and MMP-14, respectively). The results confirmed the invasive phenotype of osteosarcoma cells. Moreover, it was revealed that mRNA levels for MMP-14 differ significantly among the tested OS cell lines, suggesting that it may be an important marker of osteosarcoma development. The highest expression of MMP-14 was noted for MG-63, confirming the high invasiveness of this cell line. A potential mechanism explaining the activation of MMP-14 in MG-63 and its role during migration and invasion was presented by Wang et al. [88]. MMP-14 was identified as a crucial downstream target of the SRC/ERK1/2 signal pathway activated by WNT5A. Moreover, it was found that migration and invasion of breast cancer cells induced by BMP also correlate with increased expression of metalloproteinases, namely MMP-2, MMP-9 and MMP-14 [89]. The invasiveness of osteosarcoma cells associated with increased expression of MMP is also regulated by miRNA. Ziyan et al. showed that miR-21 affect osteosarcoma cell migration and invasion by decreasing the negative regulator of MMP, i.e., RECK (reversion-inducing-cysteine-rich protein with Kazal motifs) [90]. 

The presented gene expression profile may be a valuable complement to studies related to osteosarcomas’ cancer- and bone-related phenotypes. The classification of OS cell lines based on their transcriptome profile determined using RT-qPCR should provide support to various groups commonly applying this technique in their studies. This knowledge may be beneficial in terms of preclinical studies and the identification of a relevant model. Furthermore, an analysis of the molecular mechanism underlying osteosarcoma progression is necessary. Identifying promising therapeutic candidates may facilitate osteosarcoma diagnosis, restore drug sensitivity and improve therapeutic effects by developing more effective targeted therapies to overcome the poor prognosis of this cancer.

## 4. Materials and Methods

### 4.1. Cell Lines and Cell Culture

The cell lines, including three osteosarcoma models (Saos-2, MG 63 and U-2 OS) and human cervix epithelioid carcinoma (HeLa) were derived from the European Collection of Authenticated Cell Cultures (ECACC, Sigma-Merck, Poznan, Poland), while and human mesenchymal stromal cells (hASC) were purchased from Millipore (Sigma-Merck, Poznan, Poland). The cells were propagated at constant conditions (37 °C, 5% CO2 and 95% humidity) in sterile conditions using a CO_2_ incubator. The cells were cultured in a complete growth media consisting of basal media, i.e., McCoy’s 5A for U-2 OS and Saos-2, Eagles’ Minimum Essential Medium for MG-63 and HeLa and Dulbecco’s Modified Eagle Medium with low glucose for hASC. All media were supplemented with 10% of foetal bovine serum (FBS) and 1% of the antibiotic solution containing 10,000 units of penicillin and 10 mg of streptomycin/mL (Sigma-Merck Poznan, Poland). Cultures were maintained in T-75 flasks and harvested at logarithmic phase at confluence established at 90%. Cultures were washed in Hank’s Balanced Salt Solution without calcium and magnesium (HBSS, Sigma-Merck Poznan, Poland) and trypsinized for approximately 10 min at 37 °C (StableCell™ Trypsin Solution). Cells were centrifuged (300× *g*, 4 min), washed again, counted and homogenized with Extrazol (Blirt, Gdansk, Poland). 

### 4.2. Quantitative Reverse Transcription PCR (RT-qPCR)

All reagents used for the RT-qPCR were molecular biology grade. RNase-free plastic pipette tips, PCR tubes, microcentrifuge tubes, and conical tubes were used during the experiment to avoid degradation of the matrices. Total RNA was isolated from cell homogenates using the phenol-chloroform method published by Chomczynski and Sacchi [91]. Monolayers at a density of 1 × 10^6^ were homogenised using 1 mL of Extrazol (Blirt, Gdansk, Poland). The chloroform, isopropanol, ethanol and DEPC-treated water used during total RNA isolation were derived from Sigma-Aldrich (Poznan, Poland). Detailed protocols used to determine transcript levels were published previously [34,45,92]. In this protocol, cDNA was transcribed from 1 μg of total RNA, once purified from gDNA using DNAse I (Primerdesign, BLIRT S.A, Gdansk, Poland). The reaction of gDNA removal was performed according to the protocol provided by the producer. The cDNA synthesis was performed using Tetro cDNA Synthesis Kit (Bioline Reagents Limited, London, UK), following the manufacturer’s general protocol and oligo(dT) priming strategy. The qPCR was performed using SensiFAST SYBR^®^&Fluorescein Kit (Bioline Reagents Ltd., London, United Kingdom) in a total volume equal to 10 μL. The final concentration of primers used in the reaction was 400 nM. The primers were synthesised by Sigma-Aldrich, Poznan, Poland and purified using HPLC technology. The RT-qPCR thermal cyclers from Bio-Rad (Hercules, CA, USA) were used, i.e., T100 during RNA purification and cDNA synthesis and CFX Connect Real-Time PCR Detection System for transcript levels measurements.

The primers used for the reaction are presented in Table 1.

### 4.3. Statistical Analysis

The obtained results were verified using analysis of variance (ANOVA) with the Bonferroni correction (Bonferroni post hoc test). The analysis was performed using GraphPad Prism 8 (GraphPad Software, San Diego, CA, USA). Statistically significant differences were marked with * for *p* < 0.05, ** for *p* < 0.01, *** *p* for < 0.001 and **** *p* < 0.0001. The results are presented as mean values obtained in at least three independent reactions, while whiskers represent standard deviation (±SD).

## 5. Conclusions

In summary, we used the RT-qPCR technique to determine expression patterns of several non-coding RNAs and genes associated with OS development. We compared them with profiles obtained for the HeLa cell line and human adipose-derived stromal cells (hASCs).

We showed that miR-21-5p and BMP-7 could be relevant markers of osteosarcoma cell lines. mRNA levels for BMP receptors showed a cell-specific expression pattern. The results confirmed the molecular phenotype of the osteosarcoma cells, associated with their invasiveness and manifested by low expression of MEG3 and high mRNA levels for metalloproteinases, especially for MMP-14. Moreover, we established that the levels of miR-223 and miR-320 noted in OS cell lines and hASC are comparable. These onco-miRs could be significant players in OS development associated with the activity of ASCs in the tumour microenvironment.

The presented gene expression profile may prove to be a valuable complement to studies aimed at the characterization of osteosarcoma and its cancer- and bone-related phenotypes. Comprehensive molecular identification of OS cell lines provides helpful information that may benefit the proper selection of a model for functional analysis aimed at developing new methods of osteosarcoma treatment.

## Figures and Tables

**Figure 1 cancers-14-04533-f001:**
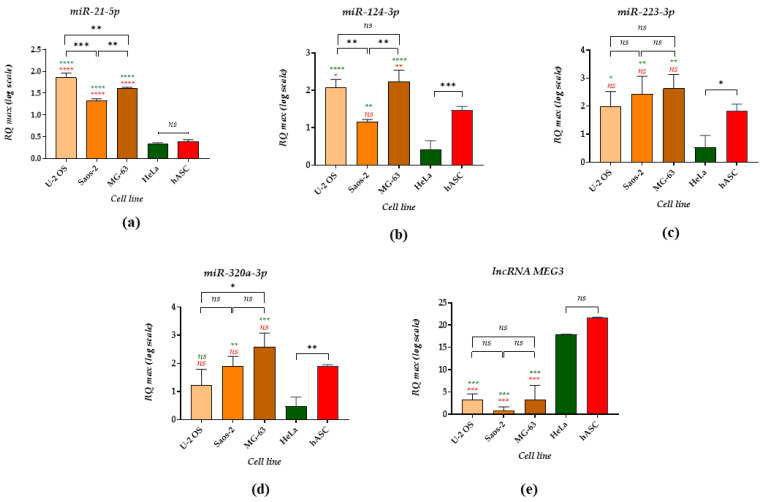
Non-coding RNA levels determined using RT-qPCR. The examined targets were small non-coding RNA (miRNA), i.e., (**a**) miR-21-5p, (**b**) miR-124-3p, (**c**) miR-223-3p, (**d**) miR-320a-3p and long-noncoding RNA (lncRNA) MEG3 (**e**). The asterisk marks statistically significant differences (* *p* < 0.05, ** *p* < 0.01, *** *p* < 0.001; **** *p*-value < 0.0001), while non-significant differences are marked as *ns*. The results of the comparative analysis are indicated using brackets. Differences in transcript levels determined between OS cell lines and reference cell lines are marked with the different coloured font—green for HeLa and red for hASC. Normalised expression of the examined targets was measured using the RQ_MAX_ method and presented on a log scale.

**Figure 2 cancers-14-04533-f002:**
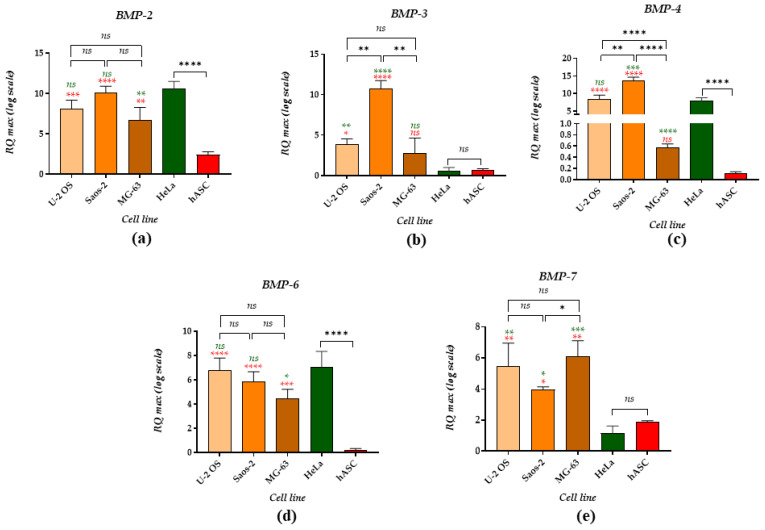
Results of mRNA levels determined for BMPs in OS cell lines and reference cells: HeLa cell line and hASC. The expression of the following transcripts was compared: (**a**) BMP-2, (**b**) BMP-3, (**c**) BMP-4, (**d**) BMP-6 and (**e**) BMP-7. The asterisk marks statistically significant differences (* *p* < 0.05, ** *p* < 0.01, *** *p* < 0.001, **** *p* < 0.0001), while non-significant differences are marked as *ns*. Transcript levels compared between cells are indicated with brackets. Differences in transcript levels determined between OS cell lines and reference cell lines are marked with different colours– green for the HeLa and red for the hASC population. The normalised expression of the examined targets was measured using the RQ_MAX_ method and presented on a log scale.

**Figure 3 cancers-14-04533-f003:**
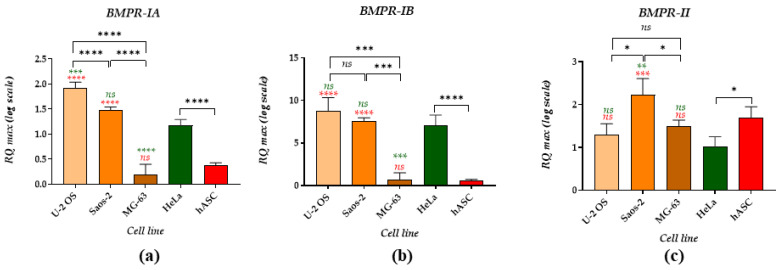
Analysis of mRNA levels of BMPRs determined in OS cell lines and reference cells: HeLa cell line, as well as hASC. The examined targets were: (**a**) BMPR-IA, (**b**) BMPR-IB, and (**c**) BMPR-II. The asterisk marks statistically significant differences (* *p* < 0.05, ** *p* < 0.01, *** *p* < 0.001, **** *p* < 0.0001), while non-significant differences are marked as *ns*. The results of the comparison of the transcripts determined between cells are indicated with brackets. Differences in transcript levels determined between OS cell lines and reference cell lines are marked with the different colours—green for the HeLa and red for the hASC population. Normalised expression of the examined targets was measured using the RQ_MAX_ method and presented on a log scale.

**Figure 4 cancers-14-04533-f004:**
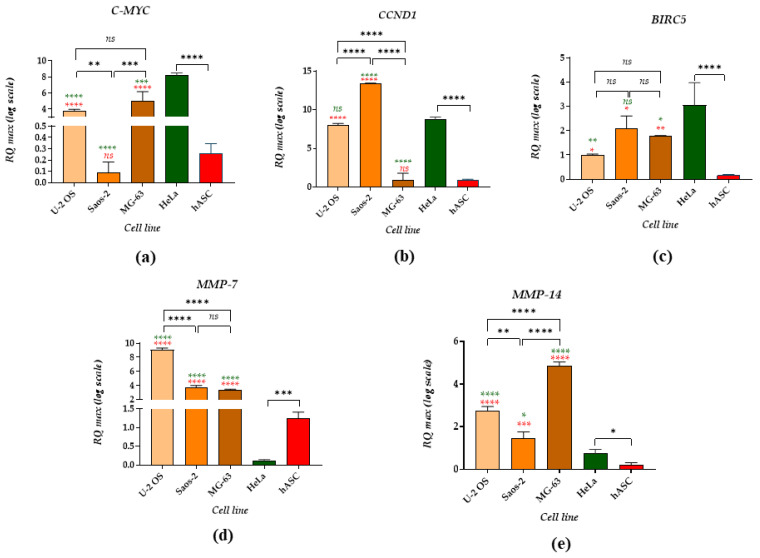
Analysis of the mRNA level of genes associated with cell cycle and viability. The examined targets were: (**a**) C-MYC, (**b**) CYCLIN D, (**c**) BIRC5, (**d**) MMP-7 and (**e**) MMP-14. The asterisk marks statistically significant differences (* *p* < 0.05, ** *p* < 0.01, *** *p* < 0.001, **** *p* < 0.0001), while non-significant differences are marked as *ns*. The results of the comparison of the transcripts determined between cells are indicated with brackets. Differences in transcript levels determined between OS cell lines and reference cell lines are marked with the different colours—green for the HeLa and red for the hASC population. Normalised expression of the examined targets was measured using the RQ_MAX_ method and presented on a log scale.

**Figure 5 cancers-14-04533-f005:**
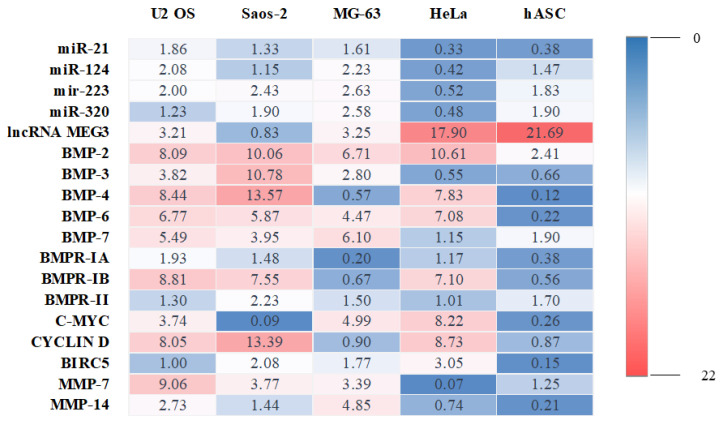
Summarized transcriptome profile noted for OS cell lines, HeLa and hASC. The colour’s intensity indicates the level of measured transcripts. Red represents up-regulation, and blue represents down-regulation.

**Table 1 cancers-14-04533-t001:** List of primers used for RT-qPCR.

Gene	Primer Sequence 5’-3’	Loci	Amplicon	Accession No.
			Length [bp]	
*BMP-2*	F: TAGACCTGTATCGCAGGCAC	1024-1043	193	NM_001200.3
	R: AACTCCTCCGTGGGGATAGA	1216-1197		
*BMP-3*	F: GGTAGACTTTGCAGATATTGGCTGG	1448-1472	148	NM_001201.3
	R: AGCTCTCACTATACTCTGGATGGTA	1595-1571		
*BMP-4*	F: CTGCTCCGGCTGAGTATCTA	181-200	192	NM_001202.5
	R: GTTGCTCGGGATGGCACTAC	372-353		
*BMP-6*	F: TCAACCGCAAGAGCCTTCTG	1402-1421	164	NM_001718.5
	R: TTTGTGGTGTCGCTGACGAG	1565-1546		
*BMP-7*	F: CAAGGCCGTCTTCAGTACCC	850-869	145	NM_001719.2
	R: CTCTCGATGGTGGTAGCGTG	994-975		
*BMPR-IA*	F: TAAAGGTGACAGTACACAGGAAACA	511-535	298	NM_004329.2
	R: TCTATGATGGCAAAGCAATGTCC	808-786		
*BMPR-IB*	F: TACAAGCCTGCCATAAGTGAGAAGC	236-258	209	NM_001203.2
	R: ATCATCGTGAAACAATATCCGTCTG	444-420		
*BMPR-II*	F: TCCTCTCATCAGCCATTTGTCCTTTC	998-1023	457	NM_001204.6
	R: AGTTACTACACATTCTTCATAG	1454-1433		
*GAPDH*	F: GTCAGTGGTGGACCTGACCT	894-913	256	NM_001289746.1
	R: CACCACCCTGTTGCTGTAGC	1149-1130		
*CCND1*	F: GATGCCAACCTCCTCAACGA	264-283	211	NM_053056.2
	R: GGAAGCGGTCCAGGTAGTTC	474-455		
*C-MYC*	F: CTTCTCTCCGTCCTCGGATTCT	1847-1868	204	NM_001354870.1
	R: GAAGGTGATCCAGACTCTGACCTT	2050-2027		
*BIRC5*	F: ACCGCATCTCTACATTCAAG	114-143	113	NM_001168.3
	R: CAAGTCTGGCTCGTTCTC	226-209		
*MMP-7*	F: TGTATGGGGAACTGCTGACA	488-507	151	NM_002423.5
	R: GCGTTCATCCTCATCGAAGT	638-619		
*MMP-14*	F: TCGGCCCAAAGCAGCAGCTTC	312-332	180	NM_004995.4
	R: CTTCATGGTGTCTGCATCAGC	491-471		
*lncRNA MEG 3*	F: GAGTGTTTCCCTCCCCAAG	474 -493	203	NR_046470.2
	R: GCGTGCCTTTGGTGATTCAG	676-657		
*miR-21a-5p*	TAGCTTATCAGACTGATGTTGA	18-39	-	MIMAT0000530
*miR-124-3p*	TAAGGCACGCGGTGAATGCC	44-63	-	MIMAT0000134
*miR-203a-3p*	GTGAAATGTTTAGGACCACTAG	65-86	-	MI0000283
*miR-320-3p*	AAAAGCTGGGTTGAGAGGGCGA	48-69	-	MI0000704

## Data Availability

The data can be shared up on request.

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
