# Peer review of "Comparison of Selected Non-Coding RNAs and Gene Expression Profiles between Common Osteosarcoma Cell Lines"

_cancers, 2022, doi:10.3390/cancers14184533_

Round 1
Reviewer 1 Report
More detailed information on long non-coding RNA MEG3 may be added in the Introduction. Figure 1 legend may be revised to indicate that MEG3 is long non-coding RNA (lncRNA).
Materials and Methods would be revised to add information on cell lines and statistics in sub-sections.
Reviewer 2 Report
The topic is interesting, the study well designed and the paper well written.
The Authors evaluated levels of several onco-miRs and long non-coding 23 RNA MEG3 in different osteosarcoma cell lines.
Aim of the study must be detailed further.
I would describe more in details overall differences between evaluated cell lines. Which are the known differences between them?
Discussion: please provide more details regarding what are the potential translations in the clinical practice. I would add a paragraph on therapies potential acting on these targets.
